# Influence of Mineral Additives on the Efflorescence of Ettringite-Rich Systems

**DOI:** 10.3390/ma14185464

**Published:** 2021-09-21

**Authors:** Linglin Xu, Siyu Liu, Peiming Wang, Zhenghong Yang

**Affiliations:** 1Key Laboratory of Advanced Civil Engineering Materials, Tongji University, Ministry of Education, Shanghai 201804, China; xulinglin@126.com (L.X.); liusiyu7288@163.com (S.L.); tjwpm@126.com (P.W.); 2School of Materials Science and Engineering, Tongji University, Shanghai 201804, China

**Keywords:** efflorescence, ettringite, mineral additives, inhibition

## Abstract

Efflorescence is aesthetically undesirable to all cementitious materials products and mainly results from the carbonation of hydrates and salt precipitation. Alternative binders without portlandite formation theoretically have much lower efflorescence risk, but in practice, the efflorescence of ettringite-rich systems is still serious. This study reports the impacts of mineral additives on the efflorescence of ettringite-rich systems and the corresponding microstructural evolution. The effects of silica fume, limestone powder, and diatomite on efflorescence and the capillary pore structure of mortars were investigated from a multi-scale analysis. The composition and microstructure of efflorescent phases were revealed by optical microscope (O.M.), in-situ Raman spectroscopy, and Scanning Electron Microscopy (SEM). Results indicate that the addition of mineral additives can efficiently inhibit the efflorescence of reference, especially with silica fume. Similar to the ettringite-rich system, the efflorescence substances of all modifies are composed of ettringite and CaCO_3_, indicating that the addition of mineral admixture does not lead to chemical reactions, lower capillary absorption coefficient of mineral additives modified specimen, the denser pore structure and the lower efflorescence degree.

## 1. Introduction

Efflorescence of cementitious materials is a serious construction quality problem due to the migration of alkali ions from the inner pore solution to the surface of mortars, concrete, or brickwork. These ions gradually react with carbon dioxide and water from the atmosphere to form whitish deposits [1,2]. Visible severe efflorescence damages the aesthetics and deteriorates the physical properties of the cementitious materials [3]. Therefore, nowadays, how to decrease the efflorescence of cementitious materials has aroused increasing attention [4,5,6].

For Portland cement, the underlying mechanisms of efflorescence involve a set of physical and chemical processes corresponding to the precipitation of white materials at the surface and carbonation due to the dissolved carbon dioxide in the pore solution [1]. Carbon dioxide would react with some hydration products, such as C-S-H and portlandite, forming calcium carbonate as described in Equations (1) and (2) [7]. It has been reported in previous studies that the efflorescence of cementitious materials is frequently detected due to excess unreacted ions remaining in the pore solution [5,8]. Therefore, the addition of calcium aluminate cement to cementitious materials converts the hydration products from C-S-H and portlandite to ettringite, which decreases the concentration of alkali ions and thus inhibiting the efflorescence [6,9]. Xue et al. [6] raised a way to inhibit efflorescence by using calcium aluminate cement as an additive. The formation of a strong cross-linked molecular structure is promoted, which results in further densification of the microstructure. Moreover, the addition of ettringite-rich admixtures and low curing temperature is proven to be effective in slowing down the rate of efflorescent [10]. Therefore, it is of great significance to investigate the ettringite-rich system, which could have a wide prospective application on the efflorescence inhibition of cementitious materials.
(1)$$3\mathrm{CaO}\cdot {2\mathrm{SiO}}_{2}\cdot 3H_{2}O+3{\mathrm{CO}}_{2}\;\to {3\mathrm{CaCO}}_{3}{+2\mathrm{SiO}}_{2}{+3H}_{2}O$$
(2)$${\mathrm{Ca}(\mathrm{OH})}_{2}{+\mathrm{CO}}_{2}\;\to {\;\mathrm{CaCO}}_{3}{+H}_{2}O$$

To inhibit the efflorescence of cementitious materials and overcome the negative factors, various methods have been investigated: (1) Considering that the efflorescence of cementitious materials is closely bound up to the reactivity of the raw materials, the ettringite-rich system with relatively light efflorescence has been chosen [11]. (2) Mineral additives such as silica fume, diatomite, and limestone powder are popularly used as pozzolans or fillers to partially replace Portland cement [12,13,14,15]. Although many studies on the impact of fly ash and silica fume on geopolymers have been investigated, they mainly focus on mechanical properties and durability, i.e., the resistance to acids and heat [16,17,18]. There is little information on the efflorescence of cementitious materials modified by mineral additives. The inhibition of efflorescence mainly starts from reducing the concentration of alkali ions and adding mineral additives could replace some cement in raw materials to reduce the ions concentration [19,20,21,22]. On the other hand, permeability is another important factor in efflorescence. Mineral additives have been widely employed in cement products to reduce permeability. Moreover, when the addition of silica fume in geopolymers is up to 15%, it could be effective in improving the resistance of efflorescence [23]. Consequently, the addition of mineral additives such as silica fume and limestone powder in cementitious materials is of great significance for efflorescence inhibition.

In recent years, mineral additives have been widely investigated to improve the mechanical properties and economic benefits of cementitious materials [24,25]. They can react with the alkali ions in the pore solution and form gels to optimizing the pore structure [9]. Given this, mineral additives including fly ash, limestone powder, and diatomite were used in this present paper to inhibit the transport properties and efflorescence of ettringite-rich systems. The impact of mineral additives on efflorescence, capillary water absorption, and microstructure were analyzed by using an optical microscope (O.M.), in situ-Raman, and Scanning Electron Microscopy (SEM), from aspects of both physical transportation and chemical formation of the efflorescence substance.

## 2. Experimental

### 2.1. Raw Materials

Commercial calcium aluminate (Imerys, Tianjin, China) and α-hemihydrate (Yongtai Gypsum Corporation, Zhengzhou, China) were employed for the preparation of ettringite-rich binder in this study. Silica fume (purity ≥ 96.72%) was obtained from the Fuhua Nano-materials Corporation (Taizhou, China). Limestone powder with a density of 2.78 g/cm^3^ was purchased from Xingyeshitai Nano-technology Corporation (Guilin, China). Class TS1 diatomite was supplied by the Julian Diatomite Corporation (Fuzhou, China). The chemical composition of mixtures was measured by X-ray fluorescence (XRF) (Eco environmental science and technology Coporation, Shanghai, China), and the result is shown in Table 1. The main mineral composition of calcium aluminate was determined by quantitative X-ray diffraction (QXRD) (Eco environmental science and technology Coporation, Shanghai, China) analysis with 20% ZnO as an internal standard. Results indicated that it contains approximately 65% CA and 35% CA_2_.

### 2.2. Mix Proportion

The mixed proportion used in this study is shown in Table 2. The ettringite-rich systems were prepared with calcium aluminate and hemihydrate, and the mineral additives were added with a concentration of 25% by mass. The sand to cement ratio was 4:1, the amount of pigment was 3% of the total amount of the binder. The quartz sand with two particle size distributions was mixed by hand, the mesh numbers are 40–70 meshes, 70–100 meshes, and the ratio of the two sands is 3:2. Additionally, the water to binder ratio of 1 was adopted in order to improve the working performance of ettringite-rich mortars, which was confirmed by JCT1024-2007.

### 2.3. Methods

#### 2.3.1. Efflorescence

The fresh mortar was poured into plastic Petri dishes with a size of *ϕ*100 mm × 5 mm. To accelerate efflorescence, samples were immersed in water. A schematic diagram of the curing methods is shown in Figure 1. Firstly, the mortar samples were prepared at 20 °C with a variation of ±1 °C for 24 h before demolding, and the relative humidity was 65 ± 10%. Moreover, a digital camera (200 D Ⅱ from Canon, Tokyo, Japan) was used to collect images of the specimen surface as primary efflorescence. Furthermore, those mortar samples were cured at 40 °C with a variation of ±1 °C for 12 h, and the relative humidity was 65 ± 10%. And the mortars were immersed in 100 mL of deionized water for 12 h. After air drying, the secondary efflorescence was recorded. A digital camera was fixed to take photographs for mortars at the same light intensity, as shown in Figure 2. To quantitatively evaluate the efflorescence degree, all those photos were processed by the Image-Pro Plus 6.0 software from Media Cybernetics^®^ (v6.0, Bethesda, MD, USA). In the software, the white pixels were selected, and the efflorescence area percentage was calculated.

#### 2.3.2. Capillary Absorption Coefficient

Capillary absorption was used to quantify the capillary porosity of mortars. The samples were firstly dried in a vacuum drying oven (DZF6020B from Subo Instrument, Shaoxing, China) at 20 °C until the mass loss was less than 0.1 g, and the liquid level should be kept 5 mm higher than the spacer, as shown in Figure 3. The amount of absorbed water was measured every 10 min in the first hour and then every hour until constant mass. The water absorption rate of the specimens and the capillary absorption coefficient of the mortars were calculated via Equations (3) and (4), respectively:(3)$$S=\delta /\sqrt{t}$$
(4)$$\delta =∆W_{t}/A$$
where S is the water absorption rate of the mortars, ∆W_t_ is the amount of water absorbed at different times (kg), A is the area of the specimen in contact with water (cm^2^), δ is the capillary absorption coefficient (kg∙m^−2^), and t is the time elapsed (h).

#### 2.3.3. Microstructure Characterization

Pastes were prepared for efflorescence characterization, with a water-to-binder ratio of 0.5, confirming the setting time requirements in GB/T 201-2015. After mixing and preparing, all these pastes were prepared with a size of 20 × 20 × 20 mm^3^ cubic mold and cured at 40 °C for 1 day and then immersed in 100 mL deionized water for 12 h, the same as the mortars. For fear of data contingency, three samples with obvious efflorescence substances of the same ratio were selected for all microstructure characterization. Before microstructural analysis (O.M., in-situ Raman, and SEM), the efflorescence substances would not be scrapped from the surface of samples, which means both the observation and analysis of efflorescence substance in this present paper are in-situ.

The efflorescence substances formed on the surface of the mortars after immersion were characterized without separation from the surface by both O.M. (ZWSP-4KCH from Advanced Micro-Fabrication Equipment, Shanghai, China) and SEM (ZEISS Sigma 300VP from Carl Zeiss, Oberkochen, Germany). The samples were a paste approximately 3 mm diameter × 1 mm thick with efflorescence substances attached to the surface. The SEM test was performed with an accelerating voltage of 2 kV and in low vacuum mode.

For a better understanding of the efflorescence mechanisms, the samples observed under the optical microscope were directly put into the Raman back-scattering conditions for the in-situ Raman spectra analysis. It was carried out with a LabRAM HR Evolution Laser Confocal Raman Microspectroscopy manufactured by HORIBA, Kyoto, Japan. The system is equipped with a 532 nm laser with an output power of 100 mW and a typical spectra from 50 to 4000 cm^−1^. To minimize the variability deriving from possible sample inhomogeneity, at least ten spectra were recorded on ten different points of each specimen area (i.e., needle-like substances, cubic-like particles).

## 3. Results and Discussion

### 3.1. Efflorescence

#### 3.1.1. Visual Observation

Table 3 presents the visual observations of primary and secondary efflorescence of different mortars. It can be seen that the primary efflorescence of the specimens is totally invisible, and its area percentage of all specimens is lower than 3%, while white deposits form slightly after immersion. Meanwhile, the secondary efflorescence of the ettringite-rich system is much more severe, and it decreases prominently with the addition of mineral admixture. This, in turn, indicates that the addition of mineral admixture greatly contributes to the inhibition of efflorescence. Among three kinds of mineral additives, silica fume modified specimen exhibits the slightest efflorescence.

#### 3.1.2. Quantitative Analysis

The efflorescence area percentage after a quantitative evaluation is listed in Table 4. For the blank specimen (ER), the primary efflorescence area percentage is 0.7%, which is the lowest degree among specimens. After being immersed in water, the secondary efflorescence area percentage of the blank specimen goes up to 85.9%. However, the secondary efflorescence area percentage presents a lower than half decrease in mineral additives modified specimens. The limestone powder mixed specimen exhibits the most severe efflorescence (35.4%), higher than other specimens with diatomite (31.1%) and silica fume (30.2%). Therefore, the addition of mineral additives can be observed to effectively inhibit efflorescence.

### 3.2. Phase Assemblage of Efflorescence Substances

#### 3.2.1. O.M.

Figure 4 shows the morphological differences of efflorescence substance on the surface of specimens. Massive needle-like and white deposits can be observed in the blank specimen (ER), while only a few clusters of white deposits can be noticed in the modified ones. For the specimen mixed with silica fume and diatomite, some granular crystals occur. In addition, the pores are filled with needle-like deposits in the specimen modified by limestone powder (LP).

#### 3.2.2. In-Situ Raman

Figure 5 shows the in-situ Raman spectra of efflorescence substances on the surface of different pastes. There are four SO_4_ Raman vibration bands of ettringite around 983 cm^−1^ and 991 cm^−1^ (v_1_ SO_4_), 452 cm^−1^ (v_2_ SO_4_), 1107 cm^−1^ (v_3_ SO_4_), and 620/628 cm^−1^ (v_4_ SO_4_). In addition, the CaCO_3_ peak at 210 cm^−1^ and 1098 cm^−1^ can also be observed in all the specimens. The efflorescence substances observed in Figure 5 (ER, point A) are mainly composed of ettringite and CaCO_3_, which exhibit a higher intensity peak of ettringite than the modified specimens. The Raman spectra of the silica fume modified specimen mostly refer to the ettringite, indicating the needle-like deposits shown in Figure 4 can be assigned to the ettringite. Previous work indicated that the CaCO_3_ resulted from the carbonation of hydrates, which is similar to the efflorescence substance of Portland cement [1]. However, the discovery of ettringite is thought-provoking to some extent. The vibration bands of modified specimens vary with different mineral additives. The band at 991 cm^−1^ (v_1_ SO_4_) becomes more and more obvious when adding diatomite (D, point A), while the band at 1098 cm^−1^ appears in the limestone powder modified cement (LP, point B).

#### 3.2.3. SEM

Figure 6 presents the morphologies of different pastes obtained by SEM. The surface of the blank specimen is covered with needle-like crystals (Point A), which are assigned to the ettringite by Raman spectrum with a standard rhombohedral particle (Point B) that falls on the network. The image of the binders modified by silica fume shows that formed ettringite is over 20 μm long (Point C), and a cube particle (Point D) is found. Combined with Raman spectrum analysis, this particle is attributed to CaCO_3_. In the pastes modified by limestone powder, coarser needle-like ettringite (Point E) appears, and the diameter of rhombohedral particle (Point F) is shorter. This seems to be one explanation for a more severe efflorescence in limestone powder modified paste. For diatomite mixed pastes, the prismatic-like ettringite (Point G) is longer compared with limestone powder mixed paste. Meanwhile, the CaCO_3_ particle (Point H) is cubic-like. The detection of ettringite and CaCO_3_ in all specimens further confirms that mineral additives inhibit efflorescence by regulating pore structure rather than changing the chemical reaction.

### 3.3. Capillary Pore Structure

A capillary absorption coefficient is a crucial tool for estimating the capillary pores and potential for ions transportation. Figure 7a,b presents the effect of mineral additives on the capillary absorption rate and capillary absorption coefficient of the ettringite-rich system, respectively. The blank specimen, which shows a higher capillary absorption capacity, exhibits the most server efflorescence. By contrast, the capillary absorption of the silica fume-modified specimen exhibits a reduction in efflorescence compared to that of the limestone powder and diatomite-modified specimens. Results reveal a direct correlation of capillary absorption to efflorescence. The specimens with lower capillary absorption exhibit a lower efflorescence.

Figure 7b shows the relationship between efflorescence area percentage and capillary absorption coefficient. Similar to the result in Figure 7a, the blank specimen exhibits the highest capillary absorption coefficient. The addition of mineral additives significantly decreases the capillary absorption coefficient, which is in greatly consistent with the results of capillary absorption. The following list of different additives is ordered from the highest capillary absorption coefficient to the lowest: limestone powder > diatomite > silica fume. Compared with the efflorescence area percentage, there is a noticeable increment in efflorescence as the capillary absorption coefficient increases. Based on the Raman spectra mentioned in Figure 5, it can be concluded that the effect of mineral admixture on the efflorescence of the ettringite-rich system is more related to the reduction in capillary absorption rather than the binding of alkalis. The higher capillary absorption coefficient of materials contributes to faster ions transportation.

### 3.4. Discussion

Based on the above-mentioned results, the efflorescence of the ettringite-rich system is visible, and the morphology of the substances can be easily observed by combining O.M. and SEM. Considering the chemical composition analysis by in-situ Raman, the efflorescence substances of ettringite-rich systems have mainly consisted of ettringite and CaCO_3_. Previous studies on the efflorescence of Portland cement were formed by the migration of portlandite from the inner pore solution to the surface of mortars, and the Ca^2+^ gradually reacts with CO_2_ and water from the atmosphere to form a white deposit of CaCO_3_ [2,26]. On the basis of the efflorescence mechanism of Portland cement and this present work, the efflorescence of the ettringite-rich system becomes obvious. As shown in Figure 8, with the progress of hydration, Ca^2+^, [Al(OH)_4_]^−^ and SO_4_^2−^ gradually diffuse in the pore solution and enrich on the surface of the binders, and then generate ettringite on the surface. Since CO_2_ from the air is continuously transmitted to the pore solution, and reacts with water to generate HCO_3_^−^ and CO_3_^2−^. Last but not least, partial ettringite is carbonated to CaCO_3_.

In terms of the effect of mineral additives on efflorescence, ref. [16] put forward that the addition of silica fume could reduce the efflorescence of Portland cement-based materials, which is consistent with our findings. In general, the following conditions contribute to the inhibition of efflorescence: (a) decreasing the number of water-soluble substances and (b) fining pore structure, preventing water from transporting to the surface [27]. Those mineral additives in the ettringite-rich system do not change the composition of efflorescence substances, but they significantly inhibit the migration of alkali ions and greatly reduce the possibility of efflorescence by improving the compactness of pore structure (especially capillary pores).

## 4. Conclusions

The impact of mineral additives (silica fume, limestone powder, and diatomite) on the efflorescence of the ettringite-rich system has been revealed from a multi-scale analysis. Some main conclusions can be drawn as follows:The incorporation of mineral additives significantly reduces the secondary efflorescence. This is attributed to the inhabitation effect of mixing mineral additives, in which silica fume is the most effective choice, followed by diatomite and limestone powder.The efflorescence substances of the ettringite-rich system are composed of CaCO_3_ and ettringite, and ettringite is the dominant phase. The inclusion of mineral additives inhibits the efflorescence without changing the chemical process.The efflorescence of ettringite-rich systems results from the ion’s transportation throughout the pores. The capillary absorption results prove that the mineral admixture can optimize the pore structure, and it is directly correlated with efflorescence. By contrast, the capillary absorption of silica fume modified specimen is the lowest, which is totally consistent with the slightest efflorescence.

## Figures and Tables

**Figure 1 materials-14-05464-f001:**
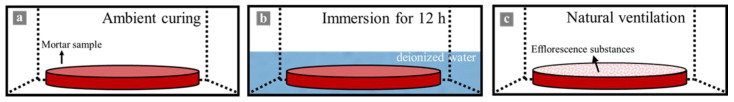
The schematic diagram of curing methods. ((**a**). Primary efflorescence; (**b**). Accelerated efflorescence by immersion; (**c**). Secondary efflorescence).

**Figure 2 materials-14-05464-f002:**
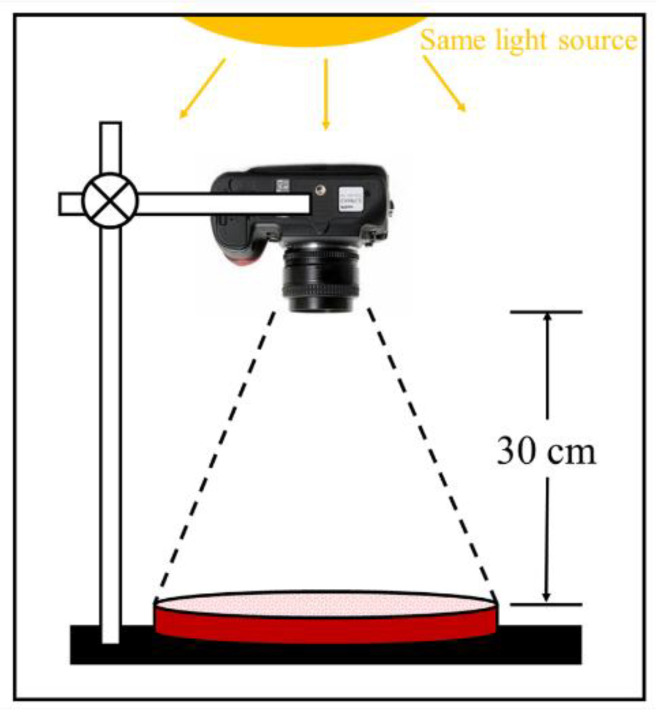
The schematic diagram of image acquisition device.

**Figure 3 materials-14-05464-f003:**
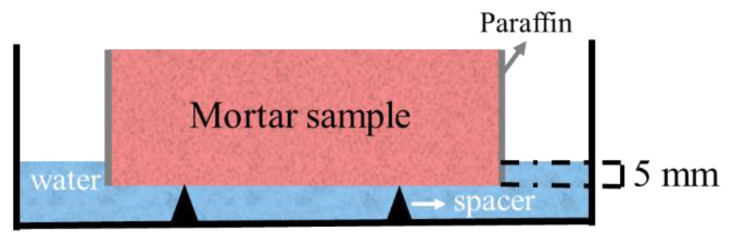
Illustration of the water absorption test.

**Figure 4 materials-14-05464-f004:**
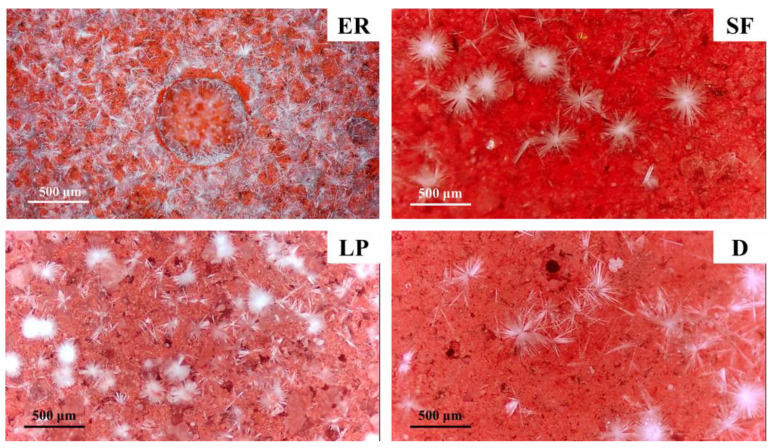
O.M. images of different specimens.

**Figure 5 materials-14-05464-f005:**
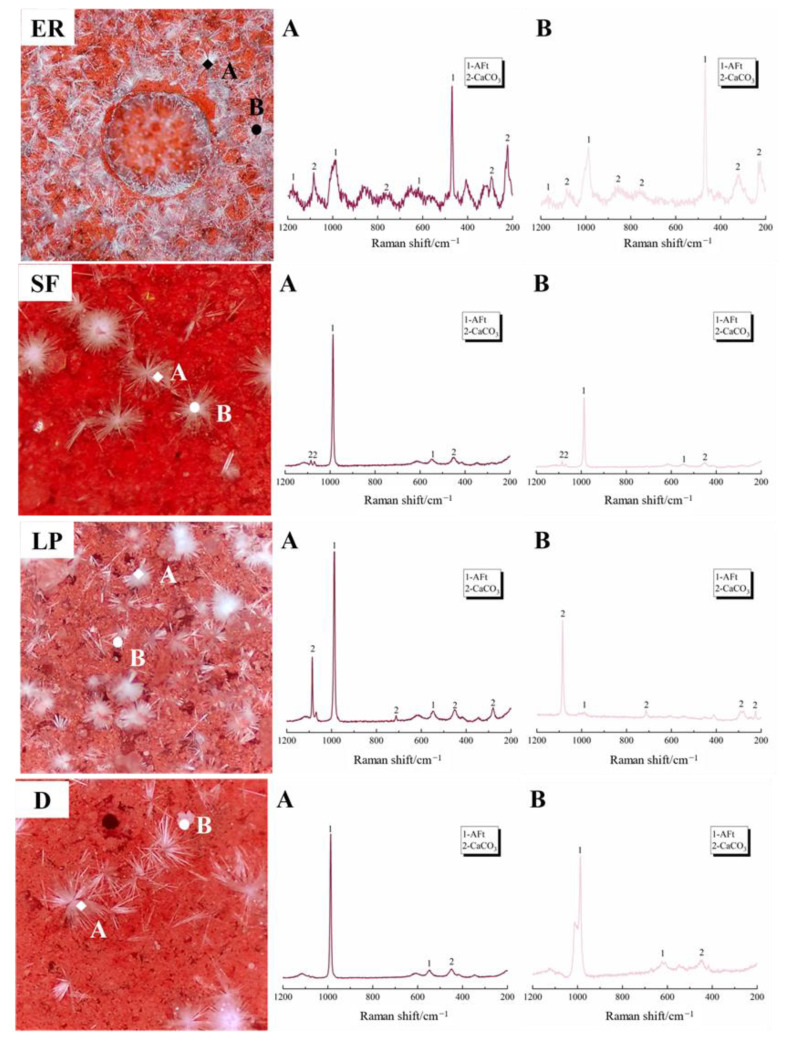
Raman spectrum of efflorescence substances on cement pastes.

**Figure 6 materials-14-05464-f006:**
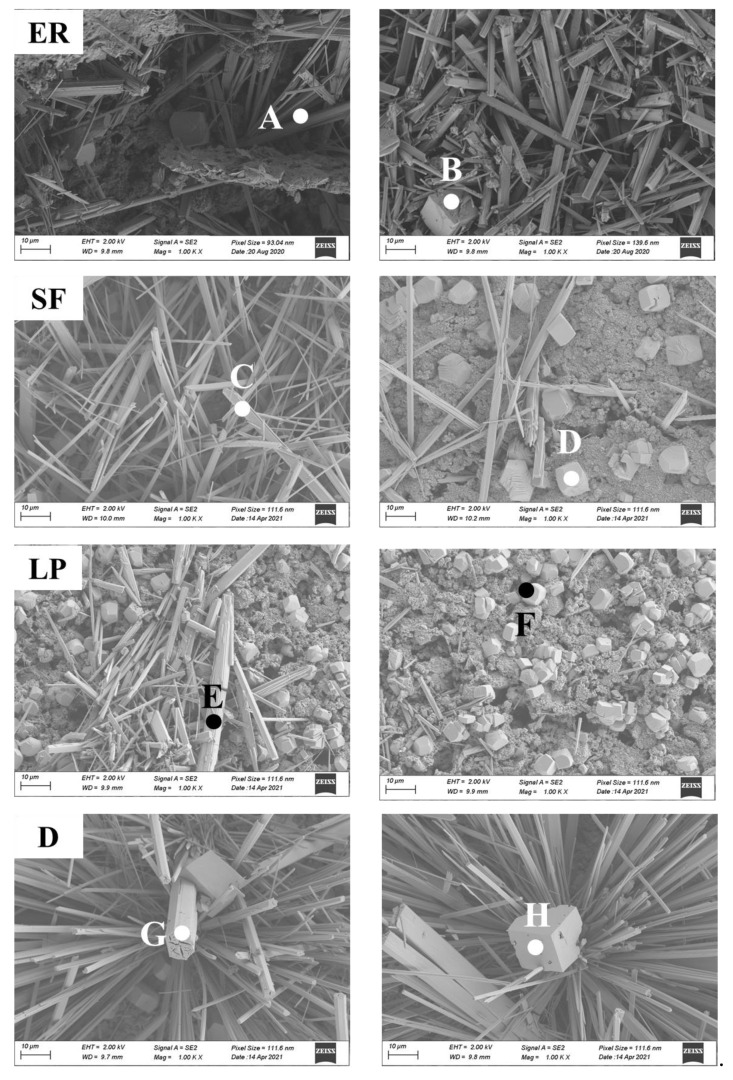
SEM images of efflorescence substances on cement pastes.

**Figure 7 materials-14-05464-f007:**
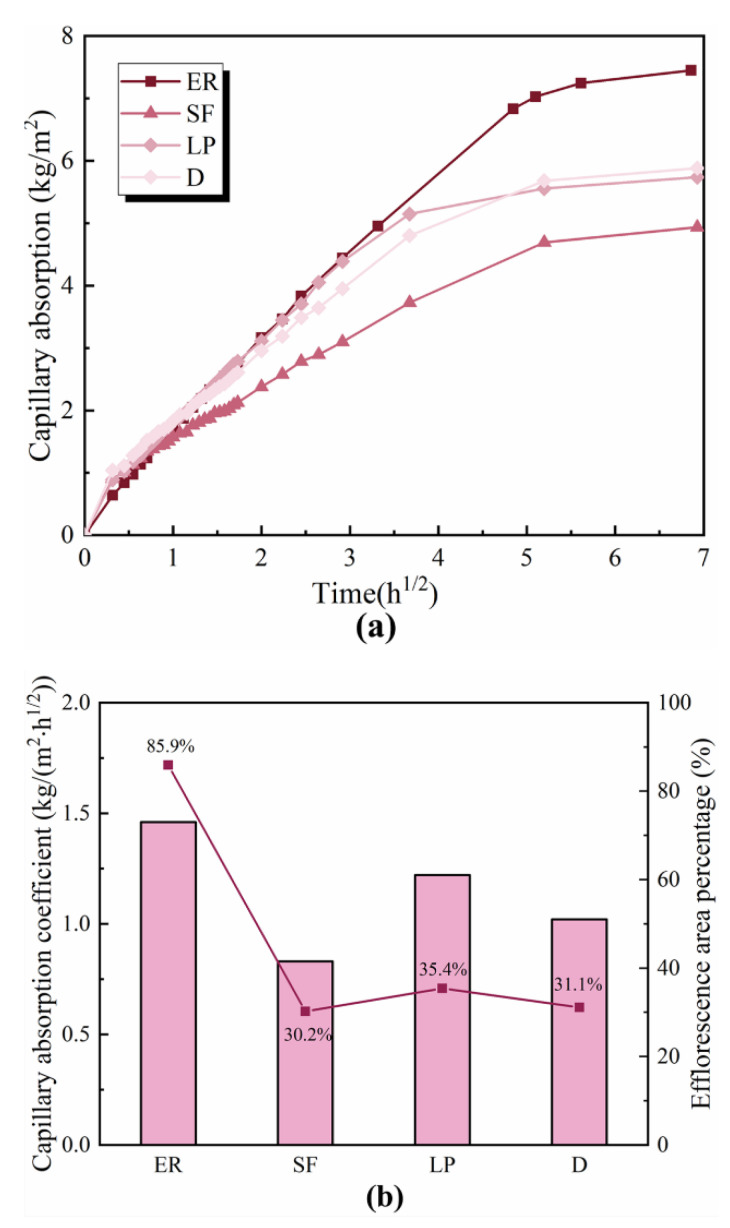
Capillary absorption (**a**) and capillary absorption coefficient (**b**) of cement mortars.

**Figure 8 materials-14-05464-f008:**
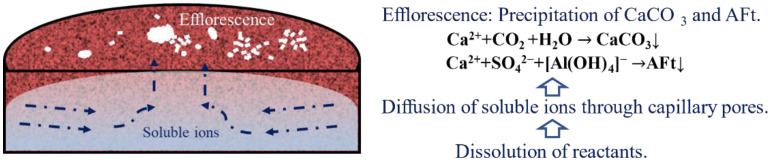
Schematic process diagram for the formation of efflorescence in ettringite-rich system.

**Table 1 materials-14-05464-t001:** Chemical composition of raw materials (wt.%).

Component	SiO_2_	CaO	Al_2_O_3_	Fe_2_O_3_	MgO	P_2_O_5_	SO_3_	K_2_O	Na_2_O	LOI
CC	0.25	30.52	68.40	0.16	0.20	-	0.08	0.01	0.32	0.06
HH	2.05	40.30	0.08	0.02	0.01	-	57.30	-	-	0.24
S	96.72	0.28	0.05	0.08	0.52	0.14	0.32	1.11	0.22	0.56
L	4.47	51.24	0.85	0.47	0.87	-	0.17	0.48	0.07	41.38
DI	94.78	0.36	2.82	0.60	0.34	0.10	-	0.39	0.19	0.42

CC: Calcium aluminate, HH: Hemihydrate, S: Silica fume, L: Limestone powder, DI: Diatomite, LOI: Loss of ignition.

**Table 2 materials-14-05464-t002:** Mix proportion of ettringite-rich systems by mass ratio.

Specimen Code	CAC	Hemihydrate	Silica Fume	Limestone Powder	Diatomite	Water	Sand	Pigment
ER	0.75	0.25	0	0	0	1	4	0.03
SF	0.5	0.25	0.25	0	0	1	4	0.03
LP	0.5	0.25	0	0.25	0	1	4	0.03
D	0.5	0.25	0	0	0.25	1	4	0.03

**Table 3 materials-14-05464-t003:** Efflorescence of cement mortars.

	ER	SF	LP	D
Primary Efflorescence	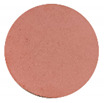	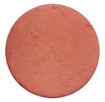	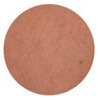	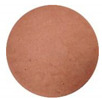
Secondary Efflorescence	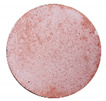	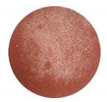	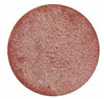	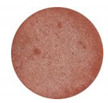

**Table 4 materials-14-05464-t004:** Efflorescence area percentage of cement mortars.

	ER	SF	LP	D
Primary Efflorescence	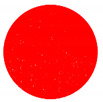	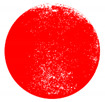	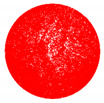	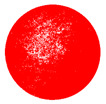
0.7%	2.4%	2.5%	3.0%
Secondary Efflorescence	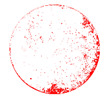	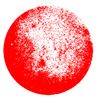	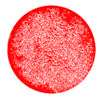	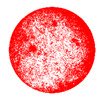
	85.9%	30.2%	35.4%	31.1%

## Data Availability

Data available on request due to restriction e.g., privacy or ethical.

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
