# Peer review of "Influence of Mineral Additives on the Efflorescence of Ettringite-Rich Systems"

_materials, 2021, doi:10.3390/ma14185464_

Round 1
Reviewer 1 Report
The paper describes the extent of efflorescence in ettringite-based mortars containing silica fume, limestone, and diatomite. The method is based on wet curing (immersion) of the mortars in water, and then taking optical images of the surface during and after immersion. The primary conclusion is that the presence of silica fume decreases porosity and mitigates efflorescence.
The suitability of the method for quantifying efflorescence needs to be explained. For instance, limestone will dissolve in water until saturation, so it not clear what the "white precipitates" on the surface represent. I also have questions about the representativeness of the SEM, Raman, and optical microscopy data. How was sampling performed? The Raman data are poorly presented, and the ability to identify specific phases only is not convincing (e.g., only 2 point measurements are conducted in each area). In general, the experiments and analyses were not performed with enough rigor. A more detailed explanation of the experimental setup used for quantifying efflorescence would also be helpful.
Author Response
We would like to thank you for your careful reading, helpful comments, and constructive suggestions, which have significantly improved the presentation of our manuscript.
Here are our responses to the comments point-by-point.
The paper describes the extent of efflorescence in ettringite-based mortars containing silica fume, limestone, and diatomite. The method is based on wet curing (immersion) of the mortars in water, and then taking optical images of the surface during and after immersion. The primary conclusion is that the presence of silica fume decreases porosity and mitigates efflorescence.
- The suitability of the method for quantifying efflorescence needs to be explained. For instance, limestone will dissolve in water until saturation, so it not clears what the "white precipitates" on the surface represent.
- Thank you so much for your valuable comment. Primary efflorescence is the white precipitates occurring during the hardening of cement-based materials. Results show that the primary efflorescence is not obvious and its area percentage of all specimens are lower than 3%. While the secondary efflorescence is the weathering of the hardened mortars, the probability of raw materials precipitation on the surface to form white deposits is relatively small. An obvious secondary efflorescence normally requires several months or years in practice. Therefore, to correctly illustrate the differences among samples and shorten the testing cycle in the laboratory, we choose the immersion method to accelerate efflorescence.
- Limestone powders themselves (without cement) can dissolve in the water, but they are integrated in the hardened specimens. Therefore, the immersion method can directly and clearly show the efflorescence degree.
- I also have questions about the representativeness of the SEM, Raman, and optical microscopy data. How was sampling performed? The Raman data are poorly presented, and the ability to identify specific phases only is not convincing (e.g., only 2 points measurements are conducted in each area). In general, the experiments and analyses were not performed with enough rigor. A more detailed explanation of the experimental setup used for quantifying efflorescence would also be helpful.
- Thank you so much for your valuable and thoughtful comment. Our experimental design was to reveal the morphology and composition of the efflorescence substance. Those zones with efflorescence were observed by naked eyes (visual observation, Table. 3) to mesoscopic (optical microscopy, Fig. 3) to microscopic (SEM, Fig. 5). According to these images, we can clearly see the substances’ morphology consist of needle-like and cubic-like particles, which means the addition of mineral additives would not change the composition of efflorescence substances.
- We chose in-situ Raman to characterize the composition of efflorescence substances mainly because the Raman spectroscopy does not require pre-processing of samples, which may give false information. We collected more than ten Raman images from one specimen, and due to the high repetition of those results, we only showed two of them in this paper.
- In general, the experiments and analyses were not performed with enough rigor. A more detailed explanation of the experimental setup used for quantifying efflorescence would also be helpful.
- We selected the immersion method to achieve the accelerated and quantitative characterization of efflorescence. The samples for efflorescence test were cylinder of 100 mm diameter and 5 mm thick. The consolidated samples were firstly exposed to 20°C for 24 h before demolding. A digital camera was used to collect images of the specimen surface as primary efflorescence. Then, the mortar samples were cured at 40°C with a variation of ± 1°C for 1 day, and the relative humidity was 65 ± 10%. Furthermore, those mortar samples were immersed in 100 mL deionized water for 12 h. After drying in air for another 12 h, the secondary efflorescence was recorded. To quantitatively evaluate the efflorescence degree, all those photos were processed through Image-Pro Plus 6.0 software. In the software, select the white pixels and calculate the efflorescence area percentage. Hope this answers your questions.
Special thanks to you for your good comments!
Reviewer 2 Report
The subject paper is interesting and its purpose complies with the journal’s aim and scope. It presents experimental results on efflorescence with the use of mineral additions. The paper is advancing the current knowledge in the field, it is original, the results are sound and adequately discussed and the manuscript is suitable for publication after minor improvements. Lastly the manuscript is generally well written and understood and minor spell check is adviced.
In greater detail, the following can be improved:
Title: better use term mineral additives or additions, not admixtures
2.2 mix proportion
Please provide a table with the exact mixtures that you have developed, showing each component with the related quantity by mass.
Line 102: preparation of mortar samples: what is the shape of the sample produced? Cube? Figure 1 shows disks. Please clarify.
Author Response
We would like to thank you for your careful reading, helpful comments, and constructive suggestions, which have significantly improved the presentation of our manuscript.
Here are our responses to the comments point-by-point.
The subject paper is interesting and its purpose complies with the journal’s aim and scope. It presents experimental results on efflorescence with the use of mineral additions. The paper is advancing the current knowledge in the field, it is original, the results are sound and adequately discussed and the manuscript is suitable for publication after minor improvements. Lastly the manuscript is generally well written and understood and minor spell check is advised.
In greater detail, the following can be improved:
- Title: better use term mineral additives or additions, not admixtures
- Thank you very much for your suggestion. We have modified the expression from mineral admixtures to mineral additives and change the expression throughout the revised manuscript.
- 2.2 mix proportion: Please provide a table with the exact mixtures that you have developed, showing each component with the related quantity by mass.
- Thank you for the reminding. We have supplemented the exact mix proportion of specimens in the corresponding position (Line 107) of the revised manuscript.
- Line 102: preparation of mortar samples: what is the shape of the sample produced? Cube?
- The mortar samples were cast into the 100 mm diameter×5 mm height plastic Petri dishes and mold results in a cylindrical sample. We explained the size of discs in the revised paper.
- Figure 1 shows disks. Please clarify.
- Thank you for the reminding. We have supplemented the description of disks (Line 110) in the manuscript.
Special thanks to you for your good comments!
Reviewer 3 Report
The article is about impact of mineral admixtures (silica fume, limestone powder and diatomite) on the efflorescence of ettringite-rich system. How ever, some issues must to be addressed:
- Please avoid bulk citations [1-6], [8-12] etc.
- References 21-25 from 2021 are reffering only to resistance on acids and heat, meanwhile your state-of-the-art are considering only 3 papers from most recent scientific literature from last 3 years. Moreover, must to be mentioned, that this kind of materials are very well studied, so introduction section must to be updated accordinlgly.
- Please check every line from table 1 if the sum of oxides =100%!
- Very poor quality of images from figure 3, 4, 5 and 6.
The article is presenting many investigations methods, but a discussion must to be inserted, where all the results to be gathered and interpreted; discussion with all correlated results is missing. Conclusion section underline some of the results explained in the paper.
Author Response
We would like to thank you for your careful reading, helpful comments, and constructive suggestions, which have significantly improved the presentation of our manuscript.
Here are our responses to the comments point-by-point.
The article is about impact of mineral admixtures (silica fume, limestone powder and diatomite) on the efflorescence of ettringite-rich system. However, some issues must to be addressed:
- Please avoid bulk citations [1-6], [8-12] etc.
- Thank you for your significant suggestion. We have corrected some repetitive references in the corresponding position of the revised manuscript. The total amount of literatures is shortened from 31 to 27.
- References 21-25 from 2021 are referring only to resistance on acids and heat, meanwhile your state-of-the-art are considering only 3 papers from most recent scientific literature from last 3 years. Moreover, must to be mentioned, that this kind of materials are very well studied, so introduction section must to be updated accordingly.
- Thank you for your significant reminding. According to your suggestion, we have updated the latest research about the mineral additives effect of mechanical properties and durability.
- Please check every line from table 1 if the sum of oxides =100%!
- We appreciate for your valuable reminding. The values of LOI have been supplemented in Table. 1 (Line 97) and now the sum of oxides reaches to 100%. Thank you so much for your careful check. This mistake has been corrected in the revised manuscript. We feel sorry for our carelessness.
- Very poor quality of images from figure 3, 4, 5 and 6.
- Thank you for the reminding. We have improved the image pixels and enlarged the size in the revised manuscript.
- The article is presenting many investigations methods, but a discussion must to be inserted, where all the results to be gathered and interpreted; discussion with all correlated results is missing. Conclusion section underline some of the results explained in the paper.
- Thank you very much for your suggestion. We have supplemented a part of discussion (3.4) in the revised manuscript.
- In this study, we tried to combine the correlation of efflorescence substances and morphology observations (OM, in-situ Raman and SEM) to reveal the efflorescence mechanism of ettringite-rich systems (Fig. 7). Generally, the mineral additives have significant suppression effects on the efflorescence of ettringite-rich systems, mainly owing to a finer capillary pore structure.
Special thanks to you for your good comments!
Reviewer 4 Report
The topic of the presented paper is interesting and definitely worth a detailed examination. Before publishing the paper, it is necessary to make a few minor adjustments and additions.
Equations (1) and (2): If water does not enter into a chemical reaction, it is not written in the chemical equation as either a reactant or a product. If the authors wanted to express the fact that the reaction takes place in an aqueous solution, the mark (aq) is written above the arrow.
Table 1: The stated chemical composition is not the chemical composition of the raw materials, as the authors claim. Limestone powder cannot contain 91.50% CaO, even pure limestone contains a maximum of 56% CaO, the remaining 44% is CO2. It would be good to supplement the table with values of L.O.I. (loss of ignition) and recalculate the oxide contents.
Caption 2.3.3: Please specify the optical microscope used.
Figure 6: The figures are not marked with pictograms a) and b).
Author Response
We would like to thank you for your careful reading, helpful comments, and constructive suggestions, which have significantly improved the presentation of our manuscript.
Here are our responses to the comments point-by-point.
The topic of the presented paper is interesting and definitely worth a detailed examination. Before publishing the paper, it is necessary to make a few minor adjustments and additions.
- Equations (1) and (2): If water does not enter into a chemical reaction, it is not written in the chemical equation as either a reactant or a product. If the authors wanted to express the fact that the reaction takes place in an aqueous solution, the mark (aq) is written above the arrow.
- Thank you for your valuable reminding. We have modified the incorrect equations in the revised manuscript. We feel sorry for our carelessness.
- Table 1: The stated chemical composition is not the chemical composition of the raw materials, as the authors claim. Limestone powder cannot contain 91.50% CaO, even pure limestone contains a maximum of 56% CaO, the remaining 44% is CO2. It would be good to supplement the table with values of L.O.I. (loss of ignition) and recalculate the oxide contents.
- Thank you so much for your suggestion. We have recalculated the oxide contents and supplemented the values of LOI in Table. 1 (Line 97). The table mistake has been corrected in the revised manuscript.
- Caption 2.3.3: Please specify the optical microscope used.
- Thank you for the reminding. We have supplemented the manufacturer and model information of optical microscope in the revised manuscript.
- Figure 6: The figures are not marked with pictograms a) and b).
- Thank you for the reminding. We have redrawn the images of Fig. 6 (a) and (b) in the corresponding position of the revised manuscript.
Round 2
Reviewer 1 Report
Some of the responses to the comments were not included/addressed in the revised manuscript, e.g., the details on the sampling protocol for the microstructural and compositional analyses. How were the samples from each mortar selected? (Were the samples scraped from the surface, 1 mm deep, etc.?) This could affect the interpretation of the results and should be documented to demonstrate sufficient rigor of the experimental methods. The number of replicate measurements was not included in the revised manuscript.
Author Response
Dear reviewers,
Thank you very much for your valuable suggestions, which are valuable in improving the quality of the paper. Following are our response on your suggestion.
The manuscript has been resubmitted to the journal. We sincerely hope that this revised manuscript has addressed all your comments and suggestions. We would like to thank you again for taking the time to review our manuscript.
Sincerely yours,
The authors

Reviewer 3 Report
The authors are completing all my requests and now, the article can be considered for publications after a severe english grammar and spelling check.
Author Response
Dear reviewer,
Thank you for your precious time on making constructive comments, which has significantly raised the quality of the manuscript and has encouraged us to enrich the manuscript. Below the comments are responded and the revisions are indicated in the latest edition.
- The authors are completing all my requests and now, the article can be considered for publications after a severe english grammar and spelling check.
- Thank you very much for your valuable reminding. We have carefully revised the manuscript according to the comments, and also have re-scrutinized to improve the English grammar and spelling check.
The manuscript has been resubmitted to the journal. We are looking forward to your positive reply. Once again, we appreciate and acknowledge your comments very much, which are valuable in improving the quality of the paper.
Sincerely yours,
The authors